# Classifying complex multimorbidity using latent class analysis and machine learning to generate insights into clustering of mental and cardiometabolic conditions

Moumita Mukherjee[1]¤*, Samhita Mukherjee[2], Hruthik Reddy Thokala[3], Raja Hashim Ali[3]

1 Wissenschaftliche Mitarbeiterin, Institute of International Health, Charité - Universitätsmedizin, Berlin, Germany, 2 Bachelor's student in Psychology, The University of Manchester, Manchester, United Kingdom, 3 Department of Business, University of Europe for Applied Sciences, Potsdam, Brandenburg, Germany

¤ Current Address: Institute of International Health, Augustenburger Platz 1, Charité – Universitätsmedizin, Berlin, Berlin, Germany
* moumita.mukherjee@charite.de

## Abstract

Machine learning techniques earn higher accuracy and robustness in multimorbidity prediction at this moment in time. Among various forms of multimorbidity, complex multimorbidity, especially the intersection of cardiometabolic disorders and mental health conditions, poses a serious threat to the public health system and needs special priority interventions. Within the scope of this context, current study aimed to define complex multimorbidity clusters using latent class analysis (LCA), test the performance of different machine learning models for accurate classification and prediction, and identify the important features by applying three feature importance techniques. The study used an excerpt of CDC Behavioral Risk Factor Surveillance System data – BRFSS 2015. It applied LCA on 46,736 responses to identify complex multimorbidity clusters and trained six machine learning algorithms (MLR, MNB, DT, RF, XGB, and ANN) in classifying the individuals falling into a typical cluster. Performance of ML models was evaluated through AUROC, accuracy, precision, recall, and F1 score. McNemar and paired T statistics are computed to find the disagreement between the ML models to verify the suitability of model selection. RF feature importance, permutation feature importance, and SHAP values are estimated to identify risk and protective factors. Five complex multimorbidity clusters emerged from LCA, dominated by mental health conditions (30% -~40%) in 1 cardiovascular cluster and 4 cardiometabolic clusters. Mental health conditions are combined with diabetes, overweight/obesity, stroke, history of heart disease, and cardiovascular risk markers. More than 60% of participants fall under complex cardiometabolic clusters who are diabetic. A greater number of overweight male/obese female with poor mental health

**Data availability statement:** Data The main data is available in https://www.cdc.gov/brfss/annual_data/annual_2014.html. The excerpt of the data available in Kaggle.com is used for analysis available in Diabetes Health Indicators Dataset https://www.kaggle.com/datasets/alexteboul/diabetes-health-indicators-dataset.

**Funding:** The APC is covered under the Charité - Universitätsmedizin Berlin publishing agreement. Our research did not receive any separate funds for research.

**Competing interests:** The authors have declared that no competing interests exist.

conditions show worse CVD markers. Random Forest model outperformed other algorithms in classification task (AUROC = 0.805, 95% CI [0.800–0.809]). Mcnemar and T statistics depict significant disagreement between the results of each ML model pair (P value = 0.0000). Feature importance analyses consistently identified age, walking difficulty, socioeconomic status, general and physical health status, education, smoking habits, physical activity status and fruit/ vegetable consumption patterns as key influencing factors. Mental health plays a critical role in shaping multi-morbidity clusters. AI-driven classification enables more accurate prediction of at-risk populations and can inform tailored interventions. This study can be considered as a use-case providing evidence for integrating ML into public health decision support.

## Introduction

### Background

Multimorbidity contributes more to premature deaths [1,2], increases hospitalization episodes, lengthens hospital stays [3], lowers quality of life, and leads to worse mental health outcomes [4,5,6]—leading to the onset of complex multimorbidity. The burden of multimorbidity is increasing globally; in a developed country, approximately 25% of the population had more than one chronic condition [7]. People with two or more conditions affecting two body systems are termed as suffering from multimorbidity, whereas people suffering from 3 or more chronic conditions are grouped under complex multimorbidity, affecting 3 or more body systems, which requires more system readiness and increases the cost of utilization [8]. The cost to the public health system has already increased due to greater requirements for visits, diagnostic tests, and sometimes lengthy hospitalization stays [6]. Besides, the multimorbid subgroups require the availability of an integrated public health care system linking respective departments with efficient coordination between different tiers of care facilities [9]. Furthermore, as indicated in a comprehensive systematic review of 44 studies role of artificial intelligence is increasing in understanding multimorbidity [10]. Therefore, system robustness is crucial to improve outcomes through accurate classification of complex multimorbid groups, ensuring the greatest number of patients receive need-based services to delay further disability and death.

Among different types of multimorbidity, cardiometabolic multimorbidity deserves significant attention in the USA, where 29.7% of individuals had it and another 29.3% were at risk [7]. Analysis of a nationally representative cross-sectional survey in the USA found one-third of survey participants had cardiometabolic multimorbidity: 14.5% with 2, 8% with 3, and 7.1% with 4 or more conditions [7]. Studying cardiometabolic multi-morbidity, particularly linkages between mental health disorders, diabetic-prediabetic conditions, and cardiovascular disorders (CVDs), falls under the complex multimorbid class affecting three body systems, warranting special attention [11,12,13].

The global burden of diabetes, CVD, and mental health issues individually contributes to significant deaths and disability worldwide. The number of patients suffering from type 2 diabetes increased from 151 million in 2000–463 million in 2019 [14];

within the age group of 20–79, 537 million people were diagnosed with diabetes in 2021, with an estimated 643 million by 2030 and 783 million by 2045 [15]. Prediabetic populations have a greater risk of diabetes, heart failure, or stroke, increasing cardiometabolic multimorbidity risk [7,16]. CVDs alone are responsible for 17.9 million deaths worldwide, accounting for 32% of all deaths, with 85% due to heart attack and stroke [17]. The WHO global action plan targets a 25% reduction in CVD and diabetes burden with 80% availability of affordable technology and infrastructure [18]. Furthermore, in 2019, 970 million people had mental health disorders, with anxiety and depression being the most prevalent and increasing [19,20]. Mental disorders result in disability for 1–6 years of life, causing 10–20 years of life lost, with both direct and indirect economic impacts [19]. Mental health issues such as anxiety, stress, posttraumatic stress disorders, and other complications are associated with endocrine and neurological damage and various chronic ailments [21,22,23,24]. Among different public health priorities, population subgroups with mental health disorders and chronic conditions affecting multiple body systems underscore the importance of timely interventions [21,22,23,24].

According to a Cochrane review, five types of interventions are evident in Smith et al. [6]: awareness generation of providers and patients for better case management, provider incentivization, interventions at the facility level to increase need-based service delivery, and interventions at the regulatory level. The study mentioned that although no intervention in a free primary care design targeting patients with multiple chronic conditions is visible, such initiatives could save enormous hospitalization costs in the future. Another group of studies attempting to develop clinical practice guidelines for people suffering from diabetes and depression reflects a gap in efficient systems to generate evidence required to create guidelines [25]. Against this backdrop, the current study focuses on exploring a novel AI-based smart decision support system using secondary data for defining and classifying several complex multimorbid subgroups to support the health system in making service delivery more targeted and equitable through robust health analytics.

## Knowledge gaps

Several studies have investigated the interplay between mental health and multimorbidity. For example, a study assessing mental health symptoms in weight gain found that depression and anxiety affect weight loss program outcomes in 238 patients [22]. A prospective experimental study by Rosenkilde et al. [21] tested the link between loneliness and the onset of T2D symptoms using data from either Danish Health and Morbidity Survey [26] or the Danish National Health Survey [27] between 2000 and 2017 which includes 465290 participants older than 16 years. A systematic review of 31 studies explored the association between heart rate variability (HRV) and psychiatric disorders, highlighting potential diagnostic utility [23]. Forte et al. [24] studied HRV variability in anxiety in a student cohort, while case–control studies examined HRV alterations in schizophrenia or autism [28].

Prucolli et al. (2023) constructed a mediation model where general mental health conditions and eating-related cognitive behaviours predicted weight loss measured by BMI changes. Rosenkilde et al. [21] applied Cox proportional hazard models to examine loneliness and T2D development. Overall, these studies indicate complex physiological-psychological interactions in multimorbid populations.

Cochrane reviews have also evaluated interventions targeting mental health among multimorbid patients. Smith et al. [6] found interventions targeting diabetes and CVD significantly reduced depression scores (SMD=−0.41; 95%), while behavioural therapy effectively reduced mental health risks in the short- (RR=1.53; 95%) and medium-term (RR=1.76; 95%) [29]. These findings emphasize that timely, targeted interventions can improve health outcomes and reduce economic burdens.

Recent studies applied machine learning (ML) techniques to classify multimorbid patients. Zaidan et al. [30] reported that random forests achieved the highest accuracy in predicting complex multimorbidity across 3-, 4-, and 5-class models, with 91–92% accuracy for diabetes+depression+CVD+hypertension combinations. A cross-sectional study with longitudinal mortality follow-up by Zheng et al. [31] applied latent class analysis (LCA) to create multimorbidity clusters in a clinically meaningful manner. A retrospective modified cross-sectional study examining sex-specific differences in cardiometabolic comorbidity performed LCA to identify distinct patient clusters with different other analyses to understand differences in adverse cardiac remodeling [32]. Study by [33] employed LCA to cluster multimorbidity pattern using the

China Health and Retirement Longitudinal Study and found four distinct morbidity patterns where all clusters show significant association with depression and predictive performance was evaluated using XGBoost model. The study by Polessa Paula et al. [34] applied different machine learning models for multimorbidity prediction. Another study by [35] developed and validated explainable ML model to predict the risk of sleep disorder among older multimorbid population subgroup followed by applying Shapley Additive Explanations to identify important features contributing to the outcome.

Despite these advances, proper clustering and classification of patients by multimorbidity severity remain underresearched in a widespread manner along with identification of most important contributing features, limiting the design of targeted interventions and resource optimization.

## Study objective

Given the increasing burden of multimorbidity, particularly complex cardiometabolic multimorbidity combined with mental health disorders, there is a critical need for AI-driven decision support systems to improve classification and risk stratification. The present study aims to explore how an AI-enabled public health decision support system can accurately classify different complex multimorbid populations and identify key determinants to optimize targeted interventions, improve health outcomes, and enhance system efficiency.

## Research questions

1. What are the best multimorbidity clusters, and what are the contributions of different components used to construct the multimorbidity phenomenon using LCA?

2. What is the best algorithm to classify the risk of having different complex multimorbidity cluster membership?

3. What are the important risk factors of developing complex multimorbidity as per feature importance analysis and SHAP?

## Materials and methods

### Dataset

The dataset is an excerpt of the 2015 Behavioral Risk Factor Surveillance System (BRFSS) from the US CDC, containing 21 features and 253,680 responses on diabetes-related health indicators, behavioral variables, health status, and demographics [36], Building Risk Prediction Models for Type 2 Diabetes Using Machine Learning Techniques. Feature engineering was performed using Stata 14.0 BE and Python 3.11 to create relevant variables and optimize classification. The dataset was loaded into Python using pandas, verified to contain no missing values, and split into input features (X) and the target variable (y).

### Data preprocessing and feature engineering

Participants who were neither prediabetic nor diabetic or had never experienced stroke were excluded, leaving 46,736 observations. Class imbalance was addressed using SMOTE to generate synthetic samples near decision boundaries. Data were split 80:20 into training and testing sets. Two new variables were created: stroke occurrence and presence/absence of mental health disorders by sex.

### Analysis

Latent class analysis (LCA) clustered participants into five complex multimorbidity classes. LCA is a probabilistic, model-based clustering method that identifies unobserved participant subgroups (classes) within a heterogeneous population based on observed responses. It estimates the probability that an individual respondent belongs to each latent class, generating class membership probabilities for each respondent.

The LCA model can be expressed as

$$[\, P(Y_i) \;=\; \{k=1\}^{\{K\}}\, k\, \{j=1\}^{\{J\}}\, P(Y_{ij} \mid C_i \;=\; k)\,]$$

Where ($Y_i$) represents observed responses for individual ($i$),
($C_i$) is the latent class,
($k$) is the prior probability of class ($k$), and
($P(Y_{ij} \mid C_i = k)$) is the conditional probability of observing response ($j$) given class ($k$).

Model selection was based on the Akaike Information Criterion (AIC) Bayesian Information Criterion (BIC). LCA allows identification of distinct clusters of different complex multimorbid respondents.

Control factors included dietary habits, lifestyle, health-seeking behavior, and socioeconomic status (Table 1).

## Machine learning models and evaluation

Six supervised algorithms were applied after classifying participants into latent classes to predict class membership and determinants: multinomial logistic regression (MLR), multinomial Naive Bayes (MNB), decision tree (DT), random forest (RF), XGBoost (XGB), and artificial neural networks (ANN). Models were evaluated using accuracy, precision, recall, F1-score, confusion matrices, and AUROC. McNemar tests assessed differences between models. Feature importance analysis, permutation analysis, and SHAP analysis ranked features by contribution, identifying key predictors. This approach enabled robust classification of complex multimorbid subgroups and identification of determinants, supporting targeted health interventions.

1. **Multinomial Logistic Regression (MLR)**: A generalized regression model for predicting categorical outcomes with more than two classes.

2. **Multinomial Naive Bayes (MNB)**: A probabilistic classifier assuming conditional independence of features given the class.

3. **Decision Tree (DT)**: A non-parametric tree-based model that splits data based on feature thresholds.

4. **Random Forest (RF)**: An ensemble of decision trees using bootstrap aggregation to reduce variance and improve predictive accuracy.

5. **Extreme Gradient Boosting (XGB)**: A boosting algorithm that sequentially combines weak learners to minimize prediction error.

6. **Artificial Neural Networks (ANN)**: Shallow machine learning model with input, hidden, and output layers that capture complex nonlinear relationships.

**Evaluation metrics.** Models were evaluated using standard classification metrics.

1. **Accuracy (Eq. 1)** denotes to what extent the classifier classifies positives.

$$\text{Accuracy} = (TP + TN)/(TP + FP + FN + TN) \tag{1}$$

2. **Precision (Eq. 2)** depicts the extent of true positive classification with respect to the total of true and false positives.

$$\text{Precision} = TP/(TP + FP) \tag{2}$$

**Table 1. Variable description – Description and coding of variables used in clustering and machine-learning models.**

| Variable name | Type | Coding Categories | Description |
|---|---|---|---|
| Diabetes_012 | Categorical | 0 = No diabetes; 1 = Prediabetes; 2 = Diabetes | Self-reported diabetes status, including prediabetes |
| HighBP | Binary | 0 = No; 1 = High blood pressure | Ever told by a health professional of having hypertension |
| HighChol | Binary | 0 = No; 1 = High cholesterol | Ever told by a health professional of having high cholesterol |
| CholCheck | Binary | 0 = No cholesterol check in last 5 years; 1 = Yes | Report of cholesterol check within past 5 years |
| BMI | Continuous | Numeric value | Body Mass Index (kg/m²) |
| Smoker | Binary | 0 = No; 1 = Yes | Ever smoked at least 100 cigarettes in lifetime |
| Stroke | Binary | 0 = No; 1 = Yes | Ever told by a health professional of having a stroke |
| HeartDiseaseorAttack | Binary | 0 = No; 1 = Yes | Coronary heart disease (CHD) or myocardial infarction (MI) |
| PhysActivity | Binary | 0 = No; 1 = Yes | Participated in physical activity in past 30 days (excluding job activity) |
| Fruits | Binary | 0 = No; 1 = Yes | Consumed fruit ≥ 1 time per day |
| Veggies | Binary | 0 = No; 1 = Yes | Consumed vegetables ≥ 1 time per day |
| HvyAlcoholConsump | Binary | 0 = No; 1 = Yes | Heavy drinking (men > 14/week, women >7/week) |
| AnyHealthcare | Binary | 0 = No; 1 = Yes | Has any form of health insurance or pre-paid health plan |
| NoDocbcCost | Binary | 0 = No; 1 = Yes | Could not see a doctor in past 12 months due to cost |
| GenHlth | Ordinal | 1 = Excellent; 2 = Very good; 3 = Good; 4 = Fair; 5 = Poor | Self-reported general health status |
| MentHlth | Continuous | 1–30 days | Number of days in past 30 days mental health was "not good" |
| PhysHlth | Continuous | 1–30 days | Number of days in past 30 days physical health was "not good" |
| DiffWalk | Binary | 0 = No; 1 = Yes | Serious difficulty walking or climbing stairs |
| Sex | Binary | 0 = Female; 1 = Male | Respondent's sex |
| Age | Categorical | 1 = 18–24; …; 9 = 60–64; 13 = 80+ | 13-level age category |
| Education | Ordinal | 1 = No schooling/kindergarten; 2 = Grades 1–8; 3 = Grades 9–11; 4 = HS/GED; 5 = Some college; 6 = College graduate | Highest educational attainment |
| Income | Ordinal | 1 = <$10,000; …; 5 = <$35,000; 8 = $75,000+ | Household income categories |

3. **Recall (Eq. 3)** is also known as sensitivity, and it measures proportion of true positives correctly classified as true positives.

$$Recall = TP/(TP + FN) \qquad (3)$$

4. **F1 score (Eq. 4)** estimates the 'harmonic mean' of two metrics – precision and recall—balancing any imbalance by giving higher weight to the lower value.

$$F1 - Score = 2 * (Precision * Recall)/(Precision + Recall) \qquad (4)$$

Where, how many cases are -

**TP (true positives),** i.e., correctly predicted as positive; **TN (true negatives),** i.e., correctly predicted as negative; **FP (false positives),** i.e., incorrectly predicted as positive; **FN (false negatives),** i.e., incorrectly predicted as negative

**Area under the receiver operating characteristic (AUROC).** Measures the ability of the model to discriminate between classes, calculated as the area under the plot of True Positive Rate vs. False Positive Rate.

**Feature importance.** To interpret model predictions, we applied random forest feature importance based on the mean decrease in impurity for each feature. - Permutation importance measures change in model performance after randomly shuffling feature values. SHAP (SHapley Additive exPlanations) quantifies the contribution of each feature to individual predictions.

**Statistical tests and decision analysis.** To validate model predictions and assess relevance, we used the following: – **Paired t-test**: Compares mean differences in continuous variables across paired samples (e.g., predicted vs. observed probabilities) to assess significant changes. - **McNemar test**: Compares classification outcomes of two correlated classifiers on the same dataset, testing if performance differs significantly. - **t-test**: For evaluating differences in continuous risk factors between classes. - **Decision curve analysis (DCA)**: Evaluates net benefit across a range of threshold probabilities, helping to identify the most useful predictive model.

These statistical tests and decision curve analyses allow us to determine whether ML models reliably distinguish between latent classes, identify significant determinants, and support decision-making for targeted interventions. This methodology aids accurate identification clustering of complex multimorbid populations while informing health systems through interpretable and reliable ML models. The workflow diagram describes the analytic steps followed in the current research (Fig 1).

## Results

### Findings from latent class analysis (LCA)

The LCA clustering is presented in Fig 2 and Table 2.

### Interpretation of disease/marker percent contribution by latent class in terms of multimorbidity

The vertical bar graph (Fig 2) demonstrates the percentage contribution of select ailments and risk factors across five latent classes. The columns provide insight into each class's composition patterns, defining each multimorbidity class. Each of the multimorbidity classes reflects co-occurrence of three or more chronic conditions. Each cluster emphasizes the dominant role of mental health status in shaping multimorbidity clusters.

Fig 3 shows the percentage of participants in different clusters.

### Class-wise interpretation

**Class 1: Complex cardiovascular multimorbidity (CVD, Stroke cluster)** – Female with moderate to frequent mental health issues contributes the most (38.2%), including the participants who had stroke, with hypertensive disorder (68.5%), high LDL cholesterol (≈60%). This class reflects a psychological and cardiovascular multimorbidity pattern, where mental health issues are linked with cardiovascular risks.

**Class 2: Mild complex cardiometabolic multimorbidity (cardiometabolic 1: diabetic CVD cluster)** – Female with mild to frequent mental health issues accounts for 63.4%, with diabetes very high (≈85%). More than 60% of the participants in this cluster have hypertension and high LDL cholesterol. This indicates a mental health–cardiometabolic multimorbidity cluster, centered on diabetes–mental health–CVD marker interactions.

**Class 3: Moderate complex cardiometabolic multimorbidity (cardiometabolic 2: diabetic, CVD, overweight cluster)** – Mental health contributes the largest share (~34%), where females with mild to frequent

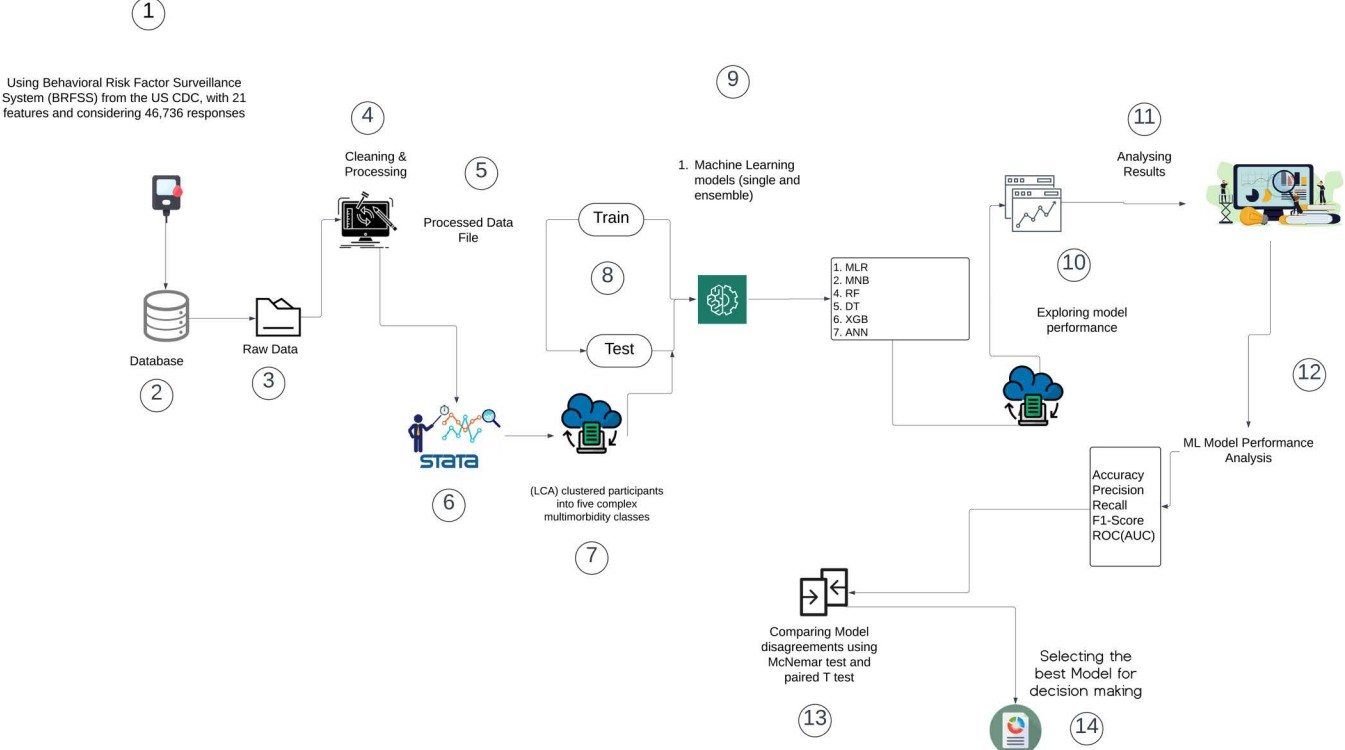

**Fig 1. The workflow diagram –** Workflow of data acquisition, preprocessing, latent class analysis, and machine-learning evaluation for classifying complex multimorbidity.

mental health issues comprise ~60% of the subgroup, with diabetes/prediabetes at ~27% (where 87% are diabetic and 13% are prediabetic), stroke at ~10%, and high blood pressure at ~10%, with 71% having hypertension. This profile can be termed as a neuro-cardiometabolic multimorbidity cluster involving mental health, diabetes, hypertension, and stroke, reflecting a high cardiovascular risk group where 100% of the overweight respondents are present.

**Class 4: Super Heightened Complex Cardiometabolic Multimorbidity (cardiometabolic3: Diabetic, CVD, Heart Disease, Obese cluster)** – Mental health is consistently significant (~30%), where ≈ 70% of females and ~ 30% of males with mild to frequent mental health conditions comprise the cluster. The contribution of diabetes/prediabetes rises to ~23%, where 89% of respondents with T2D are clustered in this subgroup. History of heart disease contributes ~10%, hypertension ~10% (including 75% with hypertensive disorder), and obesity ~10%, with 100% of respondents being obese (4301 participants are in cluster 4, who are 20% of total obese participants in the sample) in this cluster. This indicates a mixed cardiometabolic and mental health multimorbidity cluster, combining a history of vascular events and obesity-related risks.

**Class 5: Heightened Complex Cardiometabolic Multimorbidity (cardiometabolic 4: diabetic, CVD, obese cluster)** – Mental health (~33%) and diabetes/prediabetes (~29%) dominate, with more than 90% of respondents being diabetic, and the percentage of males with mild to frequent mental health issues increased in this cluster. Obesity contributes ~15%, where all the cluster 5 participants are obese (72% of the total obese of the sample), and 95% of cluster respondents have high blood pressure (contributing to ≈10%). This class illustrates a

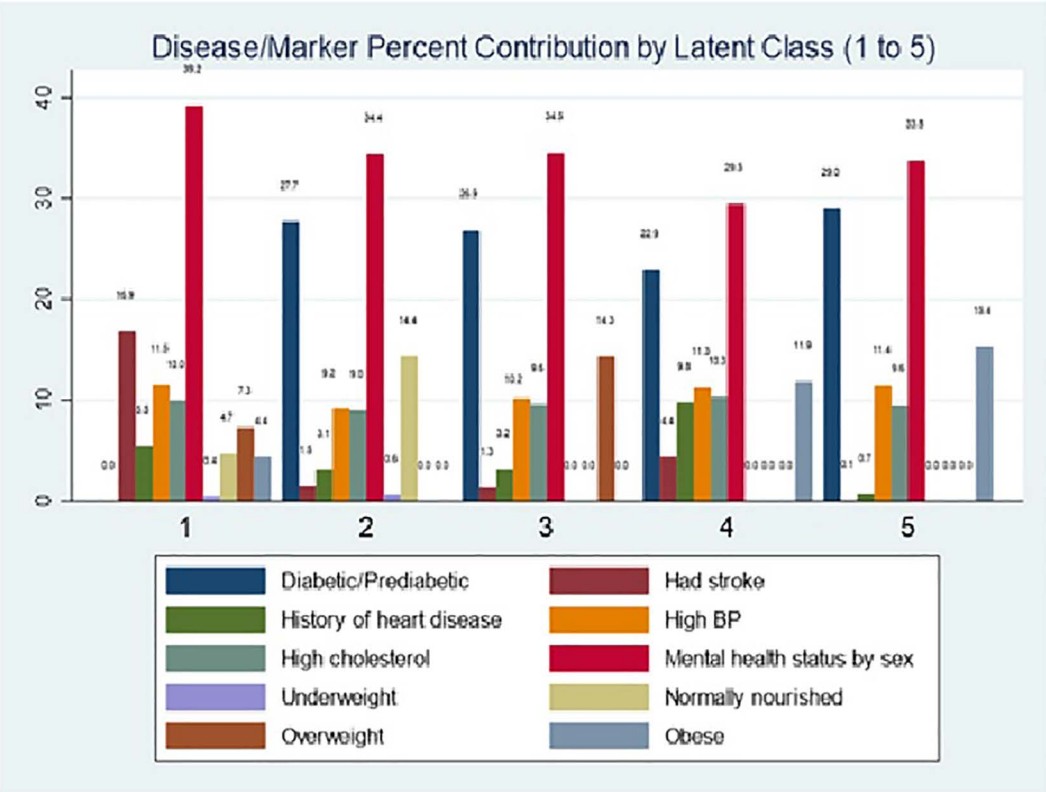

**Fig 2. Disease or biomarkers—percent contribution by latent class (class 1 to class 5)** – Distribution of disease and risk-factor contributions across the five latent multimorbidity classes identified through latent class analysis.

diabetes–obesity–mental health multimorbidity cluster, consistent with advanced cardiometabolic syndrome accompanied by psychological issues.

## Multimorbidity clusters emerged from LCA

**Mental health** appears as a dominant contributor across all the clusters. The presence of mental health issues among females clustered more in classes 1, 2, and 5, and among males in classes 3 and 4, contributing the largest proportion of ailments (29–39%), revealing its sizeable existence in multimorbidity clusters.

　**Cardiometabolic clustering:** Diabetes, hypertension, history of heart disease, stroke, obesity, and LDL cholesterol appear in different compositions and varying degrees of contributions in classes defined as four cardiometabolic clusters (class 2 to class 5), reflecting heterogeneous phenotypical multimorbidity.

　**Dual vs. complex multimorbidity:** Cardiovascular and cardiometabolic 1 clusters reflect mostly dual multimorbidity, while cardiometabolic 2–4 represent a clear presence of complex multimorbidity (≥3 conditions) if only physical morbidities are considered. Including mental health issues, which evidently play the central role, all the clusters can be termed as complex multimorbidity.

　Each cluster presents a population subgroup requiring differentiated public healthcare intervention, such as mental health–stroke integration (Class 1), diabetes–mental health management with CVD biomarker control (Class 2), diabetes–mental health management with CVD biomarker and weight control (Class 3), and comprehensive cardiometabolic–mental healthcare (Classes 4 and 5) by gender-related needs and priorities.

**Table 2. Percent distribution of each component in each latent class defining the latent class – Percent distribution of key disease markers, risk factors, and mental-health status across the five latent multimorbidity classes.**

| | | Complex cardiovascular multimorbidity | Mild complex cardiometabolic multimorbidity | Moderate complex cardiometabolic multimorbidity | Super Heightened complex cardiometabolic multimorbidity | Heightened complex cardiometabolic multimorbidity | Pr |
|---|---|---|---|---|---|---|---|
| Mental health status by gender | Female with mild mental condition | 25.3 | 27.2 | 25.6 | 20.2 | 28.7 | 0.0000 |
| | Female with moderate to frequent condition | 38.2 | 36.2 | 32.0 | 38.2 | 40.1 | |
| | Male with mild mental condition | 15.3 | 16.0 | 18.5 | 14.3 | 13.7 | |
| | Male with moderate to frequent condition | 21.3 | 20.6 | 23.9 | 27.3 | 17.5 | |
| | pre-T2D | 0.0 | 14.9 | 12.6 | 6.5 | 11.1 | 0.0000 |
| | T2D | 0.0 | 85.1 | 87.5 | 93.5 | 88.9 | |
| CVD markers | Had high BP | 68.5 | 61.5 | 71.0 | 95.1 | 74.5 | 0.0000 |
| | Had high cholesterol | 59.2 | 60.2 | 67.0 | 87.1 | 62.2 | |
| BMI | BMI < 18.5 | 2.5 | 4.1 | 0.0 | 0.0 | 0.0 | 0.0000 |
| | 18.5 < BMI < 24.5 | 28.1 | 95.9 | 0.0 | 0.0 | 0.0 | |
| | 24.5 < BMI < 30.0 | 43.5 | 0.0 | 100.0 | 0.0 | 0.0 | |
| | BMI > 30.0 | 25.9 | 0.0 | 0.0 | 100.0 | 100.0 | |
| History of heart disease/ failure | Had stroke | 100.0 | 10.0 | 9.2 | 36.9 | 0.6 | 0.0000 |
| | Had history of heart disease | 32.6 | 21.0 | 22.0 | 82.3 | 4.5 | |

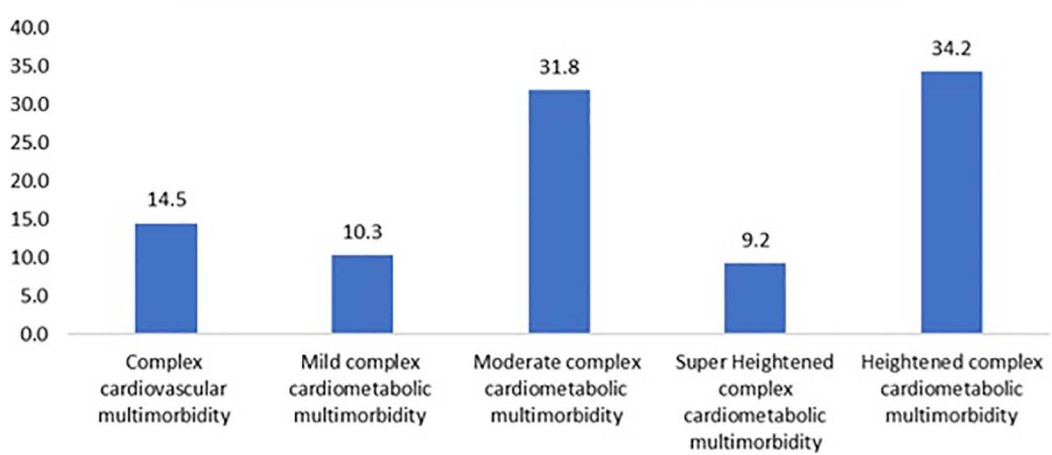

**Fig 3. Percentage distribution of participants in 5 clusters of multimorbidity (5 latent classes) –** Percentage distribution of participants within the five identified multimorbidity clusters.

## Performance of machine learning models in classifying the risk of single, multiple, and complex cardiometabolic multiple morbidities

Among 6 ML models, RF (AUROC = 0.805, 95% CI [0.800, 0.809]) outperforms all the models by model explainability (Table 3, Fig 4). XGBoost (AUROC = 0.773, 95% CI [0.769, 0.777]) is the next best model according to AUROC (OvR). The worst-performing models are the base model MLR and MNB. RF is considered the best classification algorithm while designing the DSS architecture to disaggregate each complex morbid cluster to design an equitable service delivery framework.

Pairwise model comparisons under the McNemar test reflect that each of the 6 models' performance is significantly different (p = 0.0000) from each other—indicating disagreement in model performance (Table 4).

Additionally, the results of T statistics support the findings of McNemar test results (p = 0.0000) (Table 4). The McNemar statistics, being a test of paired proportions, depict each of the 2 classifiers disagreeing with each other significantly. From the T statistics, it is also evident that when RF is compared to any other model, the T value is consistently negative while RF is the second model and positive when RF is the first model for comparison—indicating RF is significantly the best model compared to any other model and significantly different in performance.

In addition, as per decision curve analysis, RF and XGB provide higher net benefit across a wide range, implying higher usability as decision support (Fig 5).

## Discussion

In this study, we attempted to classify the participants by different types of complex multimorbidity using LCA—cardiovascular and cardiometabolic with a dominant presence of mental health conditions varying by gender in different subgroups. Then machine learning models are trained and tested to find the best algorithm in classifying these clusters with the highest accuracy and explainability, identifying significant risk and protective factors. LCA created 5 clusters of complex multimorbidity with 30% to 40% contribution of mental health condition in each cluster. Among different other chronic conditions, diabetes contributed to all 4 cardiometabolic clusters from 23% to 29%, CVD markers (9% to 11%) in all 5 clusters, contribution of overweight (14%) in cardiometabolic group 2, obesity in cardiometabolic 3 (12%) and 4 (15%), stroke (17%) in cardiovascular cluster, and history of heart disease with most contribution in cardiometabolic 3 (10%) cluster. ML models have demonstrated varying levels of effectiveness in the classification of cardiovascular to 4 cardiometabolic

**Table 3. Performance evaluation matrix: Five ML models (Base model: MLR) – Comparative performance metrics of six machine-learning algorithms in classifying complex multimorbidity clusters.**

|  | Accuracy | Precision | Recall | F1-score | Overall AUROC (OvR) | Overall AUROC CI (OvR) | Confusion Matrix |
|---|---|---|---|---|---|---|---|
| **MLR** | 0.319 | 0.309 | 0.319 | 0.305 | 0.638 | (0.633, 0.643) | [[444 824 483 940 534] [270 1086 594 668 542] [309 859 707 716 635] [280 392 411 1747 402] [256 563 498 714 1106]] |
| **MNB** | 0.301 | 0.310 | 0.301 | 0.241 | 0.619 | (0.615, 0.624) | [[185 1233 143 1478 186] [99 1624 120 1130 187] [117 1376 203 1225 305] [77 596 91 2309 159] [105 1084 157 1310 481]] |
| **Random Forest** | 0.530 | 0.538 | 0.530 | 0.532 | 0.805 | (0.800, 0.809) | [[1534 375 472 369 475] [273 1883 358 325 321] [373 319 1208 210 1116] [175 208 248 2246 355] [253 157 929 197 1601]] |
| **Decision Tree** | 0.432 | 0.433 | 0.432 | 0.432 | 0.658 | (0.653, 0.663) | [[1327 535 523 477 363] [589 1548 374 377 272] [575 414 1162 280 795] [469 429 319 1751 264] [447 310 957 305 1118]] |
| **XGBoost** | 0.457 | 0.463 | 0.457 | 0.444 | 0.773 | (0.769, 0.777) | [[665 721 593 710 536] [217 1457 524 592 370] [204 244 1285 262 1231] [202 381 295 1934 420] [71 68 853 189 1956]] |
| **ANN** | 0.348 | 0.339 | 0.348 | 0.329 | 0.678 | (0.673, 0.683) | [[498 589 487 1018 633] [336 935 545 846 498] [323 517 695 803 888] [213 206 279 1996 538] [202 278 503 715 1439]] |

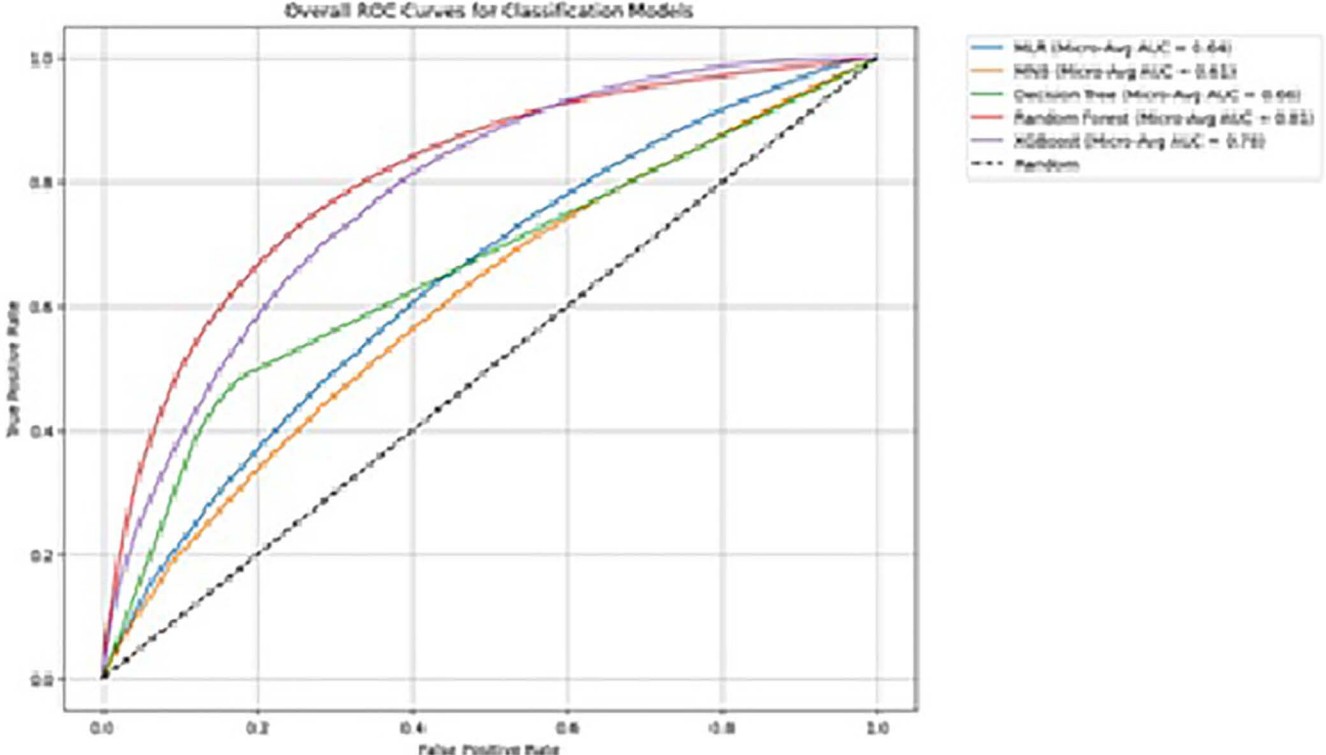

**Fig 4. AUROC curves of five ML models -** Receiver operating characteristic (ROC) curves comparing classification performance of machine-learning models.

**Table 4. McNemar test for difference in model performance – McNemar and paired t-test statistics showing pairwise differences in classification performance among machine-learning models.**

|  | McNemar Statistic | T-statistic | P-value | Significant Difference (p < 0.05) |
|---|---|---|---|---|
| MLR vs MNB | 24.5 | 157.9 | 0.0000 | TRUE |
| MLR vs Random Forest | 1755.9 | −1464.6 | 0.0000 | TRUE |
| MLR vs Decision Tree | 504.5 | −172.0 | 0.0000 | TRUE |
| MLR vs XGBoost | 1061.8 | −1232.6 | 0.0000 | TRUE |
| MLR vs ANN | 71.1 | −341.6 | 0.0000 | TRUE |
| MNB vs Random Forest | 2083.8 | −1691.1 | 0.0000 | TRUE |
| MNB vs Decision Tree | 690.4 | −342.3 | 0.0000 | TRUE |
| MNB vs XGBoost | 1147.0 | −1460.8 | 0.0000 | TRUE |
| MNB vs ANN | 141.9 | −517.3 | 0.0000 | TRUE |
| Random Forest vs Decision Tree | 615.7 | 1341.2 | 0.0000 | TRUE |
| Random Forest vs XGBoost | 286.2 | 311.5 | 0.0000 | TRUE |
| Random Forest vs ANN | 1398.9 | 1152.1 | 0.0000 | TRUE |
| Decision Tree vs XGBoost | 25.3 | −1096.2 | 0.0000 | TRUE |
| Decision Tree vs ANN | 281.4 | −176.9 | 0.0000 | TRUE |
| XGBoost vs ANN | 802.6 | 900.4 | 0.0000 | TRUE |

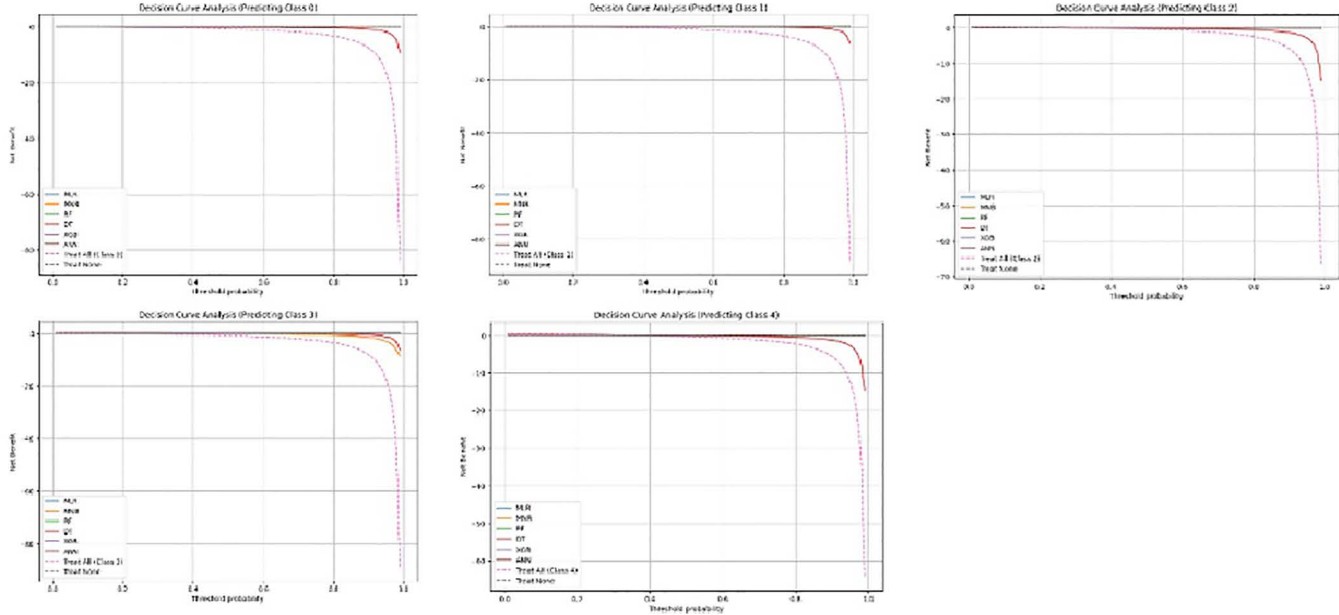

**Fig 5. Decision curves of 5 clusters** – Decision-curve analysis showing net benefit of the machine-learning models across probability thresholds for five multimorbidity classes.

complex multimorbidity classes having mild to frequent mental health conditions in a recall period of 30 days. The RF outperformed all other algorithms with the greatest accuracy and had the most balanced metrics overall. The AUROC curves in Fig 4 provide an in-depth overview of the explainability of each model, emphasizing the importance of choosing the right model in accordance with certain performance standards. Furthermore, the McNemar test and T test prove each machine learning model significantly differs from the others, confirming the best selection of 6 algorithms—no repetition in performance helps to make robust decisions on independent best model selection.

**Latent class analysis to form complex multimorbidity clusters**

**Role of mental health conditions.** Notably, the finding that the presence of perceived mental health issues significantly contributes to multimorbidity is also evident in previous studies. The study by Harshfield et al. [37] shows the influence of depression on cardiovascular disorders. In addition to this further, while looking at the mental health conditions by sex, all the clusters present a higher percentage of women with any degree of mental health conditions compared to men. A study by Kautzky et al. [38] found that after menopause, women face higher risks of cardiovascular complications. Following Kautzky et al. [38], it can be explained by considering women's health changes at the menopausal phase causing metabolic derangements. It is in concordance with studies describing the cardiometabolic changes in women increasing the risk of cardiometabolic multimorbidity [39,40,41] correlating menopause with the onset of low mood or mental health challenges amongst women [42]. Another study infers men show better adherence to lifestyle modification and management compared to women to reverse the cardiometabolic conditions – might be the reason behind the lower percentage of at-risk individuals evident among them [43].

The LCA model reflects the probability of stroke, and poor CVD markers are significantly greater among individuals with a probability of having mild to frequent mental health disorders. In cluster 2, 36% of males and 63.5% of females reflect a mild to frequent level of mental condition, among whom 85% are diabetic – is in line with Rosenkilde et al. [21], who revealed that individuals with mental health conditions and loneliness have significantly higher odds of developing

diabetes. Furthermore, the second cluster contains the highest number of prediabetic individuals (~15%), where the contribution of mental health conditions in developing mild complex cardiometabolic multimorbidity risk is 34.4%. The cohort study of [44] infers that individuals with symptoms of depression and anxiety are at higher risk of developing diabetes if they are in the prediabetic phase, implying that increasing the risk of cardiometabolic disorder supports our risk cluster 2 specification.

Besides, the third cluster depicting the risk of moderate complex cardiometabolic multimorbidity reflects that major contributors to this risk are mental health conditions (34.5%), diabetes (26.9%), overweight (14.3%), and CVD markers (~10%). Moreover, this cluster represents 57.6% of women with mild to frequent mental health conditions in the past 30 days. During the menopausal phase, women face an increased risk of CVD mortality, and the presence of psychosocial risk factors increases the risk of diabetes in women more than men [38]—in line with the risk of moderate complex cardiometabolic multimorbidity evident from LCA cluster 3.

Furthermore, clusters 3–5, depicting the risk of moderate super-heightened to heightened complex cardiometabolic multimorbidity, include contributions of ~30% to ~35% of mental health disorders, diabetes (~23% to 29%), and CVD markers (~10% to 11%), whereas overweight/obesity (12% to 15%) contributes to clusters 3 and 5, not to the super-heightened cluster. On the other hand, all the individuals in this cluster are obese, and men fall more at risk of super-heightened cluster compared to other clusters. It indicates discordance with previous studies that show higher success among men in lifestyle modification programmes with weight loss as a major component [38,45,46]. Nonetheless, those studies concentrated on cardiometabolic multimorbidity only, and the findings from the current study are different due to the mediation role of mild to frequent mental health problems.

Therefore, the current study reflects that both obese men and women with mental health issues are at risk of heightened complex cardiometabolic multimorbidity. The presence of mental health problems may work as a mediating pathway for BMI to increase the risk of cardiometabolic multimorbidity in line with previous findings. A cross-sectional study reported the mediating effects of mental conditions like emotional issues ($\beta = -0.09$, 95%), conduct behaviour ($\beta = -0.03$, 95%), and peer problems ($\beta = -0.10$, 95%) are found as the pathway of BMI influencing health-related quality of life in 313 adolescents living in a high-income setting [47]. An experimental study by Pruccoli et al. [22] in which a weight loss programme was implemented revealed that the presence of psychological disorders affects weight variability and the success elasticity of a weight loss programme varies where inhibition acts as a moderating factor [22]. According to the study, disinhibition ($\beta = -0.185$; 99%), impulsiveness ($\beta = -0.216$; 99%), perceived stress ($\beta = -0.171$; 99%), and generalized anxiety disorder ($\beta = 0.207$; 99%) are associated with impact on BMI. With that said, the limitation of the current study is the lack of detailed categories of mental health problems, which are evident in other studies.

A study by Ramesh et al. [23] revealed a significant association of HRV with phobic anxiety, panic disorder, posttraumatic stress disorder, and depression. Another study by Forte et al. [24] involved the use of student cohorts to test the associations between the variability in heart rate and trait anxiety scores. Found students with trait anxiety scores above the 80th percentile are more likely to have higher HRV than are students with the lowest 20th percentile anxiety score. Study by Haigh et al. [28] highlights future research pathways in this specific area and explores how the presence of HRVs can be used as a tool to predict physiological and psychiatric health—a major element changing multimorbidity to complex multimorbidity.

## Determinants of complex cardiovascular and cardiometabolic multimorbidity

The results of feature importance analysis after RF (Fig 6), permutation feature importance (Fig 7), and SHAP (Fig 8) revealed different rankings of factors. Feature importance analysis after RF found that older age (18%), economic status (18%), physical health status (13%), educational attainment (12%), and general health status (9%) are the major five contributing factors. Permutation importance revealed age (13%), general health (12%), facing difficulty in walking (11%),

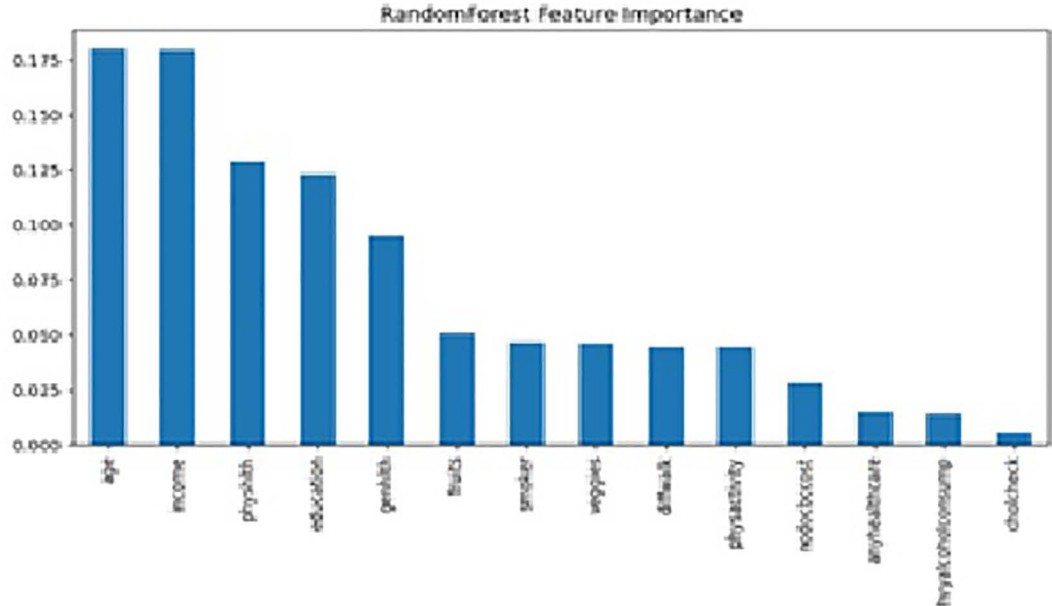

**Fig 6. Random forest feature importance –** Random-forest feature-importance ranking highlighting key predictors of complex multimorbidity.

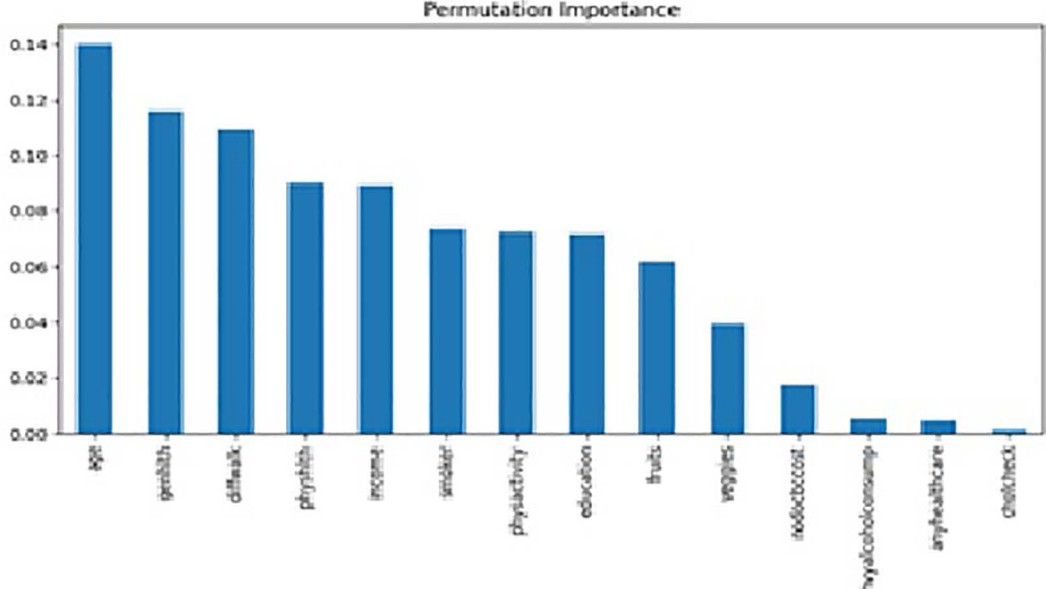

**Fig 7. Permutation feature importance –** Permutation-based feature-importance scores indicating relative contribution of each predictor variable.

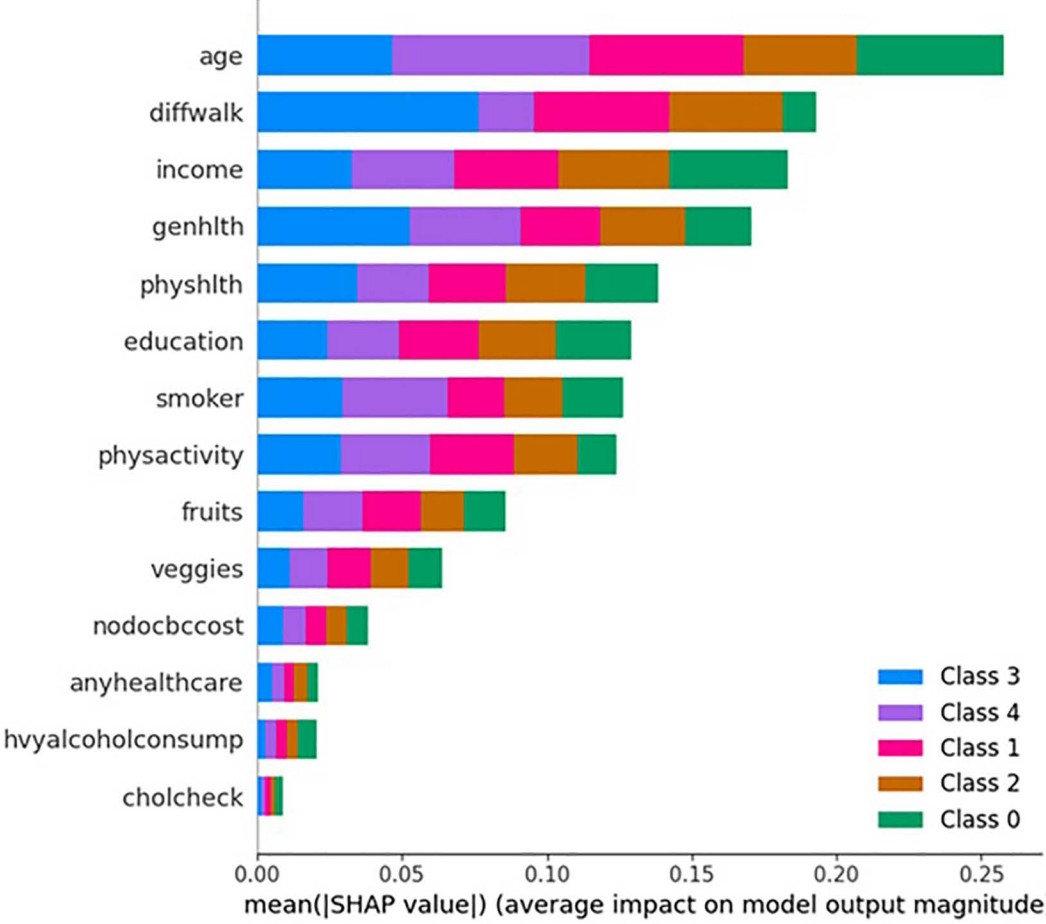

**Fig 8. SHAP – Feature importance by 5 classes created from LCA –** SHAP summary plot showing class-wise feature contributions to model predictions for multimorbidity classification.

physical health status (9%), and income level (9%) are significant contributors with great similarities between the two techniques of feature importance analysis.

However, SHAP revealed different results from feature importance after RF, and permutation feature importance. In line with the above findings, a community-based cross-sectional study by Damtie et al. [48] found older age (aOR: 6.50, 95% CI: 1.82–23.21) is significantly associated with the progression of diabetes and leads to future disease complications. Besides, cardiometabolic multimorbidity prevalence is significantly influenced by age, as is also found in Singh-Manoux et al. [49] and Chehal et al. [7]. Furthermore, like other studies, smoking habits or alcohol consumption are reflected as risk factors for this multimorbidity; however, the degree of importance of smoking is moderate (7%), but alcohol consumption is low (~1%).

In a narrative review by Joytsna et al. [50], regular physical activity and consumption of a healthy diet comprising fruits, vegetables, and lentils effectively lowers hypertension among the diabetic population—a significant pathway to reduce the cardiovascular risks in diabetic patients, which is in line with the findings that fruit and vegetable consumption habits have a 3–6% influence to reduce the risk of multimorbidity, especially among cardiovascular and the first two cardiometabolic clusters with diabetes, poor CVD markers, and some of them with a history of stroke and high BMI.

It is evident from earlier studies that the risk of progression towards cardiovascular or cardiometabolic multimorbidity along with the presence of mental health conditions is influenced by the socioeconomic status and lifestyle of individuals. Results from a cohort study show that socioeconomic and behavioural factors have a robust association with cardiometabolic multimorbidity progression above the age of 50—in line with our study [49].

## System strengthening with AI—application of machine learning for highly accurate, precise prediction

To design effective interventions and enhance the impact of the programmes accurate identification and classification of patients or at-risk populations into the above-discussed subgroups is crucial. Otherwise, equitable targeting of programmes is difficult. Additionally, understanding the complex nonlinearities among risk factors is crucial to ensure the success and elasticity of the implementation. This requires a smart system-strengthening approach. Application of artificial intelligence—machine learning algorithms to improve the predictive accuracy of introducing smart decision support—is presented in the study to explore how far it can succeed with unseen data. Therefore, this is another component in the overall novelty of this study: running machine learning algorithms and finding the best model to classify the subgroups and identify the risk factors with the highest accuracy and precision after training and testing the system—a level advancement from classical predictive modelling.

This is also in line with previous studies. Dulull et al. [51] mentioned the moderating role of advanced analytics in multimorbidity management through the generation of accurate practical insights. One study by Zaidan et al. [30] explored the best machine learning algorithm for predicting multimorbidity, including diabetes, CVD, hypertension, and depression, and reported that the random forest algorithm performed best (92% accuracy) in multiclass depression classification (4 classes) among participants with diabetes, CVD, and hypertensive disorder—in line with our study. It also achieved the highest accuracy in 3-class classification with only diabetes (91% accuracy), 5-class classification with CVD (88% accuracy) and hypertension (87% accuracy). However, a cross-sectional study to build and validate an explainable machine learning model to predict the risk of sleep disorders among older adults with multimorbidity found GBM as the best-performing model (AUROC = 0.88) among 6 ML models where RF was not applied. Nevertheless, the findings of this study are also in line with the current study, where XGBoost achieved the second-best explainability (AUROC = 0.91)—a version of GBM [35].

Based on our results, it can be inferred that the use of machine learning models can accurately classify complex multimorbidity subgroups, especially when there are borderline differences between the groups. This can be used as a cost-effective approach to help policy decisions by establishing special intervention protocol-specific public health decision support systems. As found in a previous study by Polessa Paula et al. [34] that built a similar cost-effective multimorbidity prediction model using Brazilian Longitudinal Study of Adult Health (ELSA-Brasil) data. They also found RF is the best-performing classifier in the classifier chain with 34% accuracy [34]. Public health decision support can benefit from efficient resource allocation and integration, for example, how to link psychiatric wings with cardiology and medicine and optimize skilled professionals and equipment in a customized manner following a need-based approach.

## Study novelty

Current study expands the understanding of multimorbidity by integrating latent class analysis with a comparative evaluation of several machine learning algorithms. Unlike prior research that assessed single disease dyads or attempted with a clinical objective only or relied solely on regression-based approaches, we demonstrate how mental health conditions associate with cardiometabolic disorders to form distinctly relevant multimorbidity clusters. The combined use of feature importance, permutation analysis, and SHAP further extends interpretability, offering an accurate and precise pathway for identifying high-risk subgroups. To our knowledge, this is among the first studies to embed explainable AI into the latent clustering of complex multimorbid populations in a representative dataset, emphasizing the feasibility of using such consolidative methods for public health decision support.

### Study strengths

Strengths of this study are manifold to make this study distinctive. First, the excerpt of a large and nationally representative BRFSS dataset ensures robust generalizability. Second, the application of LCA allows to identify the hidden multimorbidity subgroups in a scientific data-driven manner rather than creation of manual pre-defined clustering. Third, the comparative testing of six machine learning models, along with permutation-based and SHAP analyses, ensures thorough model validation and interpretability. Fourth, the focus on mental health as a co-occurring contributor of cardiometabolic multimorbidity adds critical insight into the biopsychosocial complexity of such chronic conditions. Finally, the study accentuates translational relevance by connecting methodological advances with their potential application in public health decision support systems, – bridging analytics with practice.

### Limitation

The current study has several strengths, as well as limitations. First, the research problem tested is based on data with limited information. More detailed information on different psychological and psychiatric biomarkers should be collected to test the main hypotheses in depth. Next, the problem considered might vary according to different ontologies and regional contexts. Therefore, the study should be followed by detailed phenomenological inductive research to identify the exact contextual constructs specific to certain regions and population subgroups. Moreover, due to the unavailability of longitudinal data, causality cannot be established. Additionally, it has found marginal variation of risk by gender, which needs further investigation not to overlook any gender equity dimension. Furthermore, it does not cover all the gaps related to clinical and public health decision support systems, as none of the authors have a clinical background. Therefore, the supply-side and institutional impacts are not captured in detail. Nevertheless, it has led to a blend of strong inferences and future research directions. Additionally, the ML approach applied in this study can be generalized to any multimorbidity prediction with a different set of chronic conditions. The study used a large sample size with a few features, ensuring a robust and feasible solution toward multimorbidity prediction.

### Future direction

The current study reflects multiple future research directions to conceptualize, build, and implement a smart AI-driven public health decision support system. An integrated health management system is to be developed to understand the epidemiology of complex multimorbidity, including multiple mental and physical chronic conditions. Advanced analytics can be applied to make robust predictions on unseen data to create multimorbidity clusters and design health intervention programmes. This would lead to the design of mental health inclusive treatment protocols considering chronic conditions related complexities to reduce the burden of multimorbidity sustainably.

### Conclusion

This study substantiates the role of mental health in cardiometabolic multimorbidity classes, highlighting how psychological conditions are interlinked with diabetes, obesity, hypertension, and cardiovascular risk. Latent class analysis resulted in robust clustering of complex multimorbidity clusters. After training and testing six ML models, Random Forest is found to be the most reliable classifier for unseen data. Feature importance analysis demonstrates the role of sociodemographic and lifestyle factors shaping the multimorbidity clustering of individuals, with age and physical status consistently emerging as leading correlates. The findings emphasize the biological, demographic, and behavioural pathways underlying complex multimorbidity, indicating the benefit of integrating AI-based decision support into public health systems. Future research would explore the feasibility of DSS to reduce misclassification cost and enable more equitable resource allocation, supporting timely intervention for vulnerable complex multimorbid subgroups. Despite data limitations and the absence of longitudinal evidence, the study offers a direction to embed advanced analytics within public health management, targeting care for patients with complex multimorbidity.

## Acknowledgments

Support received from the international health department at Charité University, offering time to the lead author to conduct research and grow in science.

## Author contributions

**Conceptualization:** Moumita Mukherjee, Raja Hashim Ali.

**Data curation:** Moumita Mukherjee.

**Formal analysis:** Moumita Mukherjee, Samhita Mukherjee.

**Investigation:** Moumita Mukherjee.

**Methodology:** Moumita Mukherjee, Raja Hashim Ali.

**Supervision:** Raja Hashim Ali.

**Validation:** Raja Hashim Ali.

**Visualization:** Moumita Mukherjee, Hruthik Reddy Thokala.

**Writing – original draft:** Moumita Mukherjee, Samhita Mukherjee.

**Writing – review & editing:** Raja Hashim Ali.

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
