## [Decision Letter · Decision Letter 0]

9 Jun 2025

Dear Dr. Mukherjee,

Thank you for submitting your manuscript to PLOS ONE. After careful consideration, we feel that it has merit but does not fully meet PLOS ONE’s publication criteria as it currently stands. Therefore, we invite you to submit a revised version of the manuscript that addresses the points raised during the review process.

We look forward to receiving your revised manuscript.

Kind regards,

Chiranjivi Adhikari, MPH, MHEd., PhD Candidate

Academic Editor

PLOS ONE

Journal Requirements:

“Not applicable”

Additional Editor Comments (if provided):

Dear Authors,

It's an interesting piece of scientific task, with well documented and reviewed reseach questions, hypothesis, and methodologies. The write up is good except with some errors and some analyses to be carried out as follows as minor revisions;

1. As also the reviewers have noted, there are errors, such as in line 158, citation 4 with ?; in 163, 95% CI missed; in 165, citation missed; in 296/299/360 and so many texts, inconsistencies between chi2 and chi-square?!

2. Similarly, some text are missing, such as line297, only 1, and 'table' is missing..

3. The table 2 shows 10 hypotheses, but below interpretation is only about three, also describe and mention the other hypotheses.

4. in line 197, write in sentence case.

5. Line 305, write in sentence case.

Followings are as major revisions:

6. For AUCs, Precision, Recall, and F1-Scores, also report 95% CIs.

7. We may only obtain the information as to how well the model ranks positive vs. negative cases, true positive rate, and other performances but limited to individual model; from AUCs, Precision, Recall, and F1-Scores. However, to compare, as for policy and clinical decision making, there should be direct comparision, which is lacking, so, head-to-head tests like DeLong’s test (for comparing AUCs statistically) and/or Bootstrap tests (to get confidence intervals on the given parameters) are recommended with consultations of statistician (senior).

8. Similarly, AUCs and other parameters as mentioned may only consider ranking, not class label predictions, and so on. Therefore, McNemar test to observe whether the number of disagreements between two classifiers is statistically significant, as with a head-to-head comparison, or similar other to compare all, are strongly advised (with statistician).

9. Similarly, the tests you have carried out may not show how performance may vary across samples or resamples, which may miss the variance and stability of performance across datasets or folds. So, for practicality, as for public health decision making, also consult for permutation tests to evaluate whether a model performs significantly better than chance or another model under label shuffling; and/or cross-validated paired t-tests or Wilcoxon signed-rank tests across folds for robust model comparison. Additionally, for better readibility, for readers who are not very comfortable with such statistics, and tests, consider the tests like decision curve analysis and Calibration plots (how well predicted probabilities reflect true likelihood).

Finally, also kindly addresss the comments from both reviewers, for which I greatly acknowledge their times.

Chiranjivi,

AE, Plos

Reviewers' comments:

Reviewer's Responses to Questions

**Comments to the Author**

1. Is the manuscript technically sound, and do the data support the conclusions?

Reviewer #1: Partly

Reviewer #2: Yes

2. Has the statistical analysis been performed appropriately and rigorously?

Reviewer #1: Yes

Reviewer #2: Yes

3. Have the authors made all data underlying the findings in their manuscript fully available?

Reviewer #1: Yes

Reviewer #2: Yes

4. Is the manuscript presented in an intelligible fashion and written in standard English?

Reviewer #1: Yes

Reviewer #2: No

Reviewer #1: The aim of this research is to explore the complex multimorbid phenomenon and to inform public health policy makers in designing smart prediction based decision making to avoid delay in specific intervention areas and increasing the targeting accuracy. Classical and machine learning models are applied to identify the best model in classifying the individuals falling in this typical subgroup to help the health system design customized solutions. However, the manuscript seems to have many limitations including the followings:

(i) There are many such works in the domain. What are the limitations of those studies? What is the significance of your work over others' works?

(ii) You should separately present the related work in a new section. Moreover, you should present a summary table for the related works.

(iii) You should have an organization paragraph at the end of your introduction section.

(iv) In the fifth page of your manuscript, you missed many references. Please check that.

(v) You should use proper reference for the dataset, not only just the Kaggle.

(vi) The quality of the images used in this manuscript are very bad.

(vii) Your material and method section is very poorly presented.

(viii) The performances of different machine learning are not satisfactory. Moreover, you have not compared your results with that of others.

(ix) Your research contributions are not satisfactory, except some analysis.

Reviewer #2: The subject of the study is quite interesting with the findings impactful. The manuscript is well written, and the methodology and research design are scientific and sound. The results are reported well, however, there are still areas of improvement in following areas:

1. Referencing should be uniform in whole papaer, "line no. 40, 46, 58, 82 96" also in others if any..You have mentioned as Rosenkilde et al. (30) in 82 and (WHO, 2020) in 96. which one is correct ? please make uniformity and follow the journal guideline

2. line no. 120 need to rewrite as " A prospective experimental study by.... tested the link between loneliness and the onset of T2D symptoms using data from the Danish National Health Survey (ref) which includes 465290 participants older than 16 years."

3. line no. 165 and 169 " . Another study by Uphoff et al. (? ) ................. explored the impact of behavioural an impact on efficacy (? ). Cannot understand reference missing or what do you want to write ? please correct.

4. line no. 185, cannot figure out the line concept. please clarify. i think the sentence structure should be in correct order.

5. line no. 297- 299 . make uniformity in the test type. Chi2 test or chi-square test.

6. regarding methodoligical choices: The study discusses various classical and machine learning methods, but it does not provide in-depth justification for the chosen methods over others and may overlook potential biases introduced by these choices. please justify.

7. table 2 seems out of page please correct it. cannot see the full table in paper layout.

8. Add one paragraph for policy implications in the future direction section which will make this papaer more compresensive

9. Conclusion: duplication must be avoided. write breifly by addressing your research objectives.

10. Add information about the strength of your study before the limitation.

Overall, the discussion section reads well.Thank you and best wishes

**Do you want your identity to be public for this peer review?** For information about this choice, including consent withdrawal, please see our Privacy Policy

Reviewer #1: No

Reviewer #2: **Yes: ** Sujan Poudel

---

## [Author Response · Author response to Decision Letter 1]

10 Oct 2025

Point by Point Response

Editor’s comments

As also the reviewers have noted, there are errors, such as in line 158, citation 4 with ?; in 163, 95% CI missed; in 165, citation missed; in 296/299/360 and so many texts, inconsistencies between chi2 and chi-square Corrected

Similarly, some text are missing, such as line297, only 1, and 'table' is missing Texts are corrected as per changes in the analysis

The table 2 shows 10 hypotheses, but below interpretation is only about three, also describe and mention the other hypotheses.

These analyses are omitted, and new methods are applied. Research questions are now strengthened. The new analysis – LCA followed by machine learning – do not include previous hypotheses.

in line 197, write in sentence case. Line 305, write in sentence case. Sentences are now changed.

For AUCs, Precision, Recall, and F1-Scores, also report 95% CIs. Computed and added

We may only obtain the information as to how well the model ranks positive vs. negative cases, true positive rate, and other performances but limited to individual model; from AUCs, Precision, Recall, and F1-Scores. However, to compare, as for policy and clinical decision making, there should be direct comparision, which is lacking, so, head-to-head tests like DeLong’s test (for comparing AUCs statistically) and/or Bootstrap tests (to get confidence intervals on the given parameters) are recommended with consultations of statistician (senior). Similarly, AUCs and other parameters as mentioned may only consider ranking, not class label predictions, and so on. Therefore, McNemar test to observe whether the number of disagreements between two classifiers is statistically significant, as with a head-to-head comparison, or similar other to compare all, are strongly advised (with statistician). We added McNemar test and paired T test to observe whether the number of disagreements between two classifiers is statistically significant following the recommendations.

So, for practicality, as for public health decision making, also consult for permutation tests to evaluate whether a model performs significantly better than chance or another model under label shuffling; and/or cross-validated paired t-tests or Wilcoxon signed-rank tests across folds for robust model comparison. Additionally, for better readibility, for readers who are not very comfortable with such statistics, and tests, consider the tests like decision curve analysis and Calibration plots (how well predicted probabilities reflect true likelihood). Yes, we have computed paired T test and Decision curve analysis and added in the new version.

Reviewer 1

There are many such works in the domain. What are the limitations of those studies? We have modified our focus to complex multimorbidity. We applied latent class analysis to define the clusters. Then we applied different ML algorithms to explore the best algorithm for classification. We considered previous works like

1. Polessa Paula, D., Barbosa Aguiar, O., Pruner Marques, L., Bensenor, I., Suemoto, C. K., Mendes da Fonseca, M. J., & Griep, R. H. (2022). Comparing machine learning algorithms for multimorbidity prediction:An example from the Elsa-Brasil study. PloS one, 17(10), e0275619.https://doi.org/10.1371/journal.pone.0275619

2. Wang, X., Zheng, N., & Yin, M. (2025a). Multimorbidity Patterns and Depression: Bridging Epidemiological Associations with Predictive Analytics for Risk Stratification. Healthcare (Basel, Switzerland), 13(12), 1458. https://doi.org/10.3390/healthcare13121458

3. Wang, X., Zhang, D., Lu, L., Meng, S., Li, Y., Zhang, R., Zhou, J., Yu, Q., Zeng, L., Zhao, J., Zeng, Y., & Gao, R. (2025b). Development and validation of an explainable machine learning model for predicting the risk of sleep disorders in older adults with multimorbidity: a cross-sectional study. Frontiers in public health, 13, 1619406.https://doi.org/10.3389/fpubh.2025.1619406

4. Bertrand, A., Zhou, X., Lewis, A., Monfeuga, T., Gupta, R., Grau, V., & Rodriguez, B. (2025). Sex-specific cardiometabolic multimorbidity, metabolic syndrome and left ventricular function in heart failure with preserved ejection fraction in the UK Biobank. Cardiovascular diabetology, 24(1), 238. https://doi.org/10.1186/s12933-025-02788-4

What is the significance of your work over others' works?

You should separately present the related work in a new section. Moreover, you should present a summary table for the related works.

You should have an organization paragraph at the end of your introduction section.

Recent studies applied machine learning (ML) techniques to classify multimorbid patients. Zaidan et al. (2023) reported that random forests achieved the highest accuracy in predicting complex multimorbidity across 3-, 4-, and 5-class models, with 91–92% accuracy for diabetes + depression + CVD + hypertension combinations. A cross-sectional study with longitudinal mortality follow-up by Zhang et al. (2021) applied latent class analysis (LCA) to create multimorbidity clusters in a clinically meaningful manner A retrospective modified cross-sectional study examining sex-specific differences in cardiometabolic comorbidity performed LCA to identify distinct patient clusters with different other analyses to understand differences in adverse cardiac remodeling (Bertrand et al., 2025). Study by Wang et al. (2025a) employed LCA to cluster multimorbidity patter using the China Health and Retirement Longitudinal Study and found four distinct morbidity patterns where all clusters show significant association with depression and predictive performance was evaluated using XGBoost model. The study by Polessa Paula et al. (2022) applied different machine learning models for multimorbidity prediction. Another study by Wang et al. (2025b) developed and validated explainable ML model to predict the risk of sleep disorder among older multimorbid population subgroup followed by applying Shapley Additive Explanations to identify important features contributing to the outcome.

Despite these advances, proper clustering and classification of patients by multimorbidity severity remain under-researched in an widespread manner along with identification of most important contributing features, limiting the design of targeted interventions and resource optimization.

In the fifth page of your manuscript, you missed many references. Please check that. Corrected.

You should use proper reference for the dataset, not only just the Kaggle. Added.

The main data is available in

https://www.cdc.gov/brfss/annual_data/annual_2014.html

The excerpt of the data available in Kaggle.com is used for analysis available in

Diabetes Health Indicators Dataset

The quality of the images used in this manuscript are very bad. Our sincere apologies. Now we are sharing the MS Word version with modified analyses and new figures. Hope you like them.

Your material and method section is very poorly presented. Our sincere apologies again. We wrote it differently improving the quality of writing.

Materials and Methods

Dataset: The dataset is an excerpt of the 2015 Behavioral Risk Factor Surveillance System (BRFSS) from the US CDC, containing 21 features and 253,680 responses on diabetes-related health indicators, behavioral variables, health status, and demographics (Xie et al., 2019), Building Risk Prediction Models for Type 2 Diabetes Using Machine Learning Techniques. Feature engineering was performed using Stata 14.0 BE and Python 3.11 to create relevant variables and optimize classification. The dataset was loaded into Python using pandas, verified to contain no missing values, and split into input features (X) and the target variable (y).

Data Preprocessing and Feature Engineering: Participants who were neither prediabetic nor diabetic or had never experienced stroke were excluded, leaving 46,736 observations. Class imbalance was addressed using SMOTE to generate synthetic samples near decision boundaries. Data were split 80:20 into training and testing sets. Two new variables were created: stroke occurrence and presence/absence of mental health disorders by sex.

Analysis: Latent class analysis (LCA) clustered participants into five complex multimorbidity classes. LCA is a probabilistic, model-based clustering method that identifies unobserved participant subgroups (classes) within a heterogeneous population based on observed responses. It estimates the probability that an individual respondent belongs to each latent class, generating class membership probabilities for each respondent.

The LCA model can be expressed as [ P(Y_i) = {k=1}^{K} k {j=1}^{J} P(Y{ij} | C_i = k) ]

Where (Y_i) represents observed responses for individual (i),

(C_i) is the latent class,

(k) is the prior probability of class (k), and

(P(Y{ij} | C_i = k)) is the conditional probability of observing response (j) given class (k).

Model selection was based on the Akaike Information Criterion (AIC) Bayesian Information Criterion (BIC). LCA allows identification of distinct clusters of different complex multimorbid respondents.

Control factors included dietary habits, lifestyle, health-seeking behavior, and socioeconomic status (Table 1).

Table 1 here

Machine Learning Models and Evaluation: Six supervised algorithms were applied after classifying participants into latent classes to predict class membership and determinants: multinomial logistic regression (MLR), multinomial Naive Bayes (MNB), decision tree (DT), random forest (RF), XGBoost (XGB), and artificial neural networks (ANN). Models were evaluated using accuracy, precision, recall, F1-score, confusion matrices, and AUROC. McNemar tests assessed differences between models. Feature importance analysis, permutation analysis, and SHAP analysis ranked features by contribution, identifying key predictors. This approach enabled robust classification of complex multimorbid subgroups and identification of determinants, supporting targeted health interventions.

1. Multinomial Logistic Regression (MLR): A generalized regression model for predicting categorical outcomes with more than two classes.

2. Multinomial Naive Bayes (MNB): A probabilistic classifier assuming conditional independence of features given the class.

3. Decision Tree (DT): A non-parametric tree-based model that splits data based on feature thresholds.

4. Random Forest (RF): An ensemble of decision trees using bootstrap aggregation to reduce variance and improve predictive accuracy.

5. Extreme Gradient Boosting (XGB): A boosting algorithm that sequentially combines weak learners to minimize prediction error.

6. Artificial Neural Networks (ANN): Shallow machine learning model with input, hidden, and output layers that capture complex nonlinear relationships.

Evaluation Metrics: Models were evaluated using standard classification metrics.

1. Accuracy denotes to what extent the classifier classifies positives.

Accuracy = (TP + TN) / (TP + FP + FN + TN) ………………………. Eq (1)

2. Precision depicts the extent of true positive classification with respect to the total of true and false positives.

Precision = TP / (TP + FP) ……………………………………………... Eq (2)

3. Recall is also known as sensitivity, and it measures proportion of true positives correctly classified as true positives.

Recall = TP / (TP + FN) ……………………………………………….. Eq (3)

4. F1 score estimates the ‘harmonic mean’ of two metrics - precision and recall—balancing any imbalance by giving higher weight to the lower value.

F1-Score = 2 * (Precision * Recall) / (Precision + Recall) …………….. Eq (4)

Where, how many cases are -

TP (true positives), i.e., correctly predicted as positive; TN (true negatives), i.e., correctly predicted as negative; FP (false positives), i.e., incorrectly predicted as positive; FN (false negatives), i.e., incorrectly predicted as negative

Area Under the Receiver Operating Characteristic (AUROC): Measures the ability of the model to discriminate between classes, calculated as the area under the plot of True Positive Rate vs. False Positive Rate.

Feature Importance: To interpret model predictions, we applied random forest feature importance based on the mean decrease in impurity for each feature. - Permutation importance measures change in model performance after randomly shuffling feature values. SHAP (SHapley Additive exPlanations) quantifies the contribution of each feature to individual predictions.

Statistical Tests and Decision Analysis: To validate model predictions and assess relevance, we used the following: - Paired t-test: Compares mean differences in continuous variables across paired samples (e.g., predicted vs. observed probabilities) to assess significant changes. - McNemar test: Compares classification outcomes of two correlated classifiers on the same dataset, testing if performance differs significantly. - t-test: For evaluating differences in continuous risk factors between classes. - Decision curve analysis (DCA): Evaluates net benefit across a range of threshold probabilities, helping to identify the most useful predictive model.

These statistical tests and decision curve analyses allow us to determine whether ML models reliably distinguish between latent classes, identify significant determinants, and support decision-making for targeted interventions. This methodology aids accurate identification clustering of complex multimorbid populations while informing health systems through interpretable and reliable ML models.

The performances of different machine learning are not satisfactory. Your research contributions are not satisfactory, except some analysis.

Please find below the new results –

Performance of machine learning models in classifying the risk of single, multiple, and complex cardiometabolic multiple morbidities

Table 2 here

Figure 4 here

Among 6 ML models, RF (AUROC=0.805, 95% CI [0.800, 0.809]) outperforms all the models by model explainability (Table 2, Figure 4). XGBoost (AUROC=0.773, 95% CI [0.769, 0.777]) is the next best model according to AUROC (OvR). The worst-performing models are the base model MLR and MNB. RF is considered the best classification algorithm while designing the DSS architecture to disaggregate each complex morbid cluster to design an equitable service delivery framework.

Table 3 here

Pairwise model comparisons under the McNemar test reflect that each of the 6 models’ performance is significantly different (p=0.0000) from each other—indicating disagreement in model performance (Table 3).

Additionally, the results of T statistics support the findings of McNemar test results (p=0.0000) (Table 3). The McNemar statistics, being a test of paired proportions, depict each of the 2 classifiers disagreeing with each other significantly. From the T statistics, it is also evident that when RF is compared to any other model, the T value is consistently negative while RF is the second model and positive when RF is the first model for comparison—indicating RF is significantly the best model compared to any other model and significantly different in performance.

Table 4 here

In addition, as per decision curve analysis, RF and XGB provide higher net benefit across a wide range, implying higher usability as decision support (Table 4).

Moreover, you have not compared your results with that of others. It is now added in detail in the discussion section with different subsections

Reviewer 2

Referencing should be uniform in whole papaer, "line no. 40, 46, 58, 82 96" also in others if any..You have mentioned as Rosenkilde et al. (30) in 82 and (WHO, 2020) in 96. which one is correct ? please make uniformity and follow the journal guideline Apologies again. Corrected.

line no. 120 need to rewrite as " A prospective experimental study by.... tested the link between loneliness and the onset of T2D symptoms using data from the Danish National Health Survey (ref) which includes 465290 participants older than 16 years." Added as “A prospective experimental study by Rosenkilde et al. (2024) tested the link between loneliness and the onset of T2D symptoms using data from either Danish Health and Morbidity Survey (Jensen et al., 2019) or the Danish National Health Survey (Christensen et al., 2022) between 2000 and 2017 which includes 465290 participants

---

## [Editor Report · Decision Letter 1]

15 Oct 2025

Classifying Complex Multimorbidity Using Latent Class Analysis and Machine Learning to Generate Insights into Clustering of Mental and Cardiometabolic Conditions

PONE-D-24-56938R1

Dear Dr. Moumita Mukherjee,

We’re pleased to inform you that your manuscript has been judged scientifically suitable for publication and will be formally accepted for publication once it meets all outstanding technical requirements.

Kind regards,

Chiranjivi Adhikari, MPH, MHEd., PhD Candidate

Academic Editor

PLOS ONE
---

## [Editor Report · Acceptance letter]

PONE-D-24-56938R1

PLOS ONE

Dear Dr. Mukherjee,

I'm pleased to inform you that your manuscript has been deemed suitable for publication in PLOS ONE. Congratulations! Your manuscript is now being handed over to our production team.

Kind regards,

on behalf of

Mr. Chiranjivi Adhikari

Academic Editor

PLOS ONE